# Promoting H$_2$O$_2$ production via 2-electron oxygen reduction by coordinating partially oxidized Pd with defect carbon

Qiaowan Chang[1], Pu Zhang [1], Amir Hassan Bagherzadeh Mostaghimi[2], Xueru Zhao[3], Steven R. Denny[4], Ji Hoon Lee[4], Hongpeng Gao[1], Ying Zhang[5], Huolin L. Xin [6 ✉], Samira Siahrostami [2 ✉], Jingguang G. Chen [3,4 ✉] & Zheng Chen [1,7 ✉]

Electrochemical synthesis of H$_2$O$_2$ through a selective two-electron (2e$^−$) oxygen reduction reaction (ORR) is an attractive alternative to the industrial anthraquinone oxidation method, as it allows decentralized H$_2$O$_2$ production. Herein, we report that the synergistic interaction between partially oxidized palladium (Pd$^{\delta+}$) and oxygen-functionalized carbon can promote 2e$^−$ ORR in acidic electrolytes. An electrocatalyst synthesized by solution deposition of amorphous Pd$^{\delta+}$ clusters (Pd$_3{}^{\delta+}$ and Pd$_4{}^{\delta+}$) onto mildly oxidized carbon nanotubes (Pd$^{\delta+}$-OCNT) shows nearly 100% selectivity toward H$_2$O$_2$ and a positive shift of ORR onset potential by ~320 mV compared with the OCNT substrate. A high mass activity (1.946 A mg$^{-1}$ at 0.45 V) of Pd$^{\delta+}$-OCNT is achieved. Extended X-ray absorption fine structure characterization and density functional theory calculations suggest that the interaction between Pd clusters and the nearby oxygen-containing functional groups is key for the high selectivity and activity for 2e$^−$ ORR.

[1] Department of NanoEngineering, University of California San Diego, La Jolla, California 92093, USA. [2] Department of Chemistry, University of Calgary, 2500 University Drive NW, Calgary, Alberta, Canada T2N 1N4. [3] Chemistry Division, Brookhaven National Laboratory, Upton, New York 11973, USA. [4] Department of Chemical Engineering, Columbia University, New York, New York 10027, USA. [5] School of Mineral Processing and Bioengineering, Central South University, Changsha 410083, China. [6] Department of Physics and Astronomy, University of California, Irvine, California 94720, USA. [7] Program of Chemical Engineering, University of California San Diego, La Jolla, California 92093, USA. ✉email: huolinx@uci.edu; samira.siahrostami@ucalgary.ca; jgchen@columbia.edu; zhengchen@eng.ucsd.edu

Hydrogen peroxide ($H_2O_2$) is one of the most important chemicals[1] that is widely used in fiber and paper production, chemical synthesis, wastewater treatment, and the mining industry[2–4]. Today's anthraquinone oxidation-based industrial production of $H_2O_2$ needs to be improved to significantly reduce energy consumption and organic waste generation[5]. In addition, the chemical instability of $H_2O_2$ poses safety issues for transportation and storage. In practice, dilute $H_2O_2$ solution suffices for most applications[6] (e.g., <0.1 wt. % $H_2O_2$ aqueous solution is used for water treatment)[7]. To enable on-demand, decentralized production of $H_2O_2$ using renewable electricity[6,8–12], electrochemical $H_2O_2$ synthesis through a selective 2-electron (2e−) oxygen reduction reaction (ORR) pathway stands out as a promising alternative route. The key to realize this process on a large scale is to develop efficient and economically viable electrocatalysts with high selectivity and activity.

In alkaline and neutral electrolytes, defective carbon materials, such as oxidized carbon nanotubes (O-CNTs)[13], B-N-doped carbon[14], Fe single-atom coordinated O-CNT[11], and reduced graphene oxide (GO)[15], have shown high activity and selectivity for the 2e− ORR. For example, mildly reduced GO exhibits nearly 100% selectivity and stable activity at low overpotential (<10 mV) in 0.1 M KOH[9,15]. It is particularly interesting to find that the selectivity of carbon materials could also be enhanced by the introduction of boron nitride (BN) islands where the active sites were attributed to the interface between hexagonal BN and graphene[14]. Although these catalysts are efficient in alkaline conditions, producing $H_2O_2$ under acidic conditions shows technological advantage in fuel cell operation as today's proton conducting polymeric membranes are far more technologically mature than their hydroxide-conducting counterparts[6,16]. In addition, acidic $H_2O_2$ solution can be directly used as an oxidant for chemical synthesis, which contributes more than 33% to the global market share of $H_2O_2$[10,17]. Due to the weak acidic nature of the $H_2O_2$ molecule[18], storing $H_2O_2$ in an acidic environmental can also offer a longer shelf-life compared to alkaline conditions. However, carbon-based materials require a large overpotential (~300 mV) to initiate the ORR reaction in acidic electrolytes, resulting in significant voltage loss in fuel cell operations[19,20]. For instance, the onset potential of high-selectivity mesoporous N-doped carbon was up to ~0.5 V in 0.1 M $HClO_4$, leading to a possible potential loss of ~200 mV in the ORR test[19].

Precious metals and alloys have long been investigated as electrocatalysts for 2e− ORR in the acidic environment, including Au[21], Pt[22,23], Pd-Au[24,25], Pt-Hg[8], Ag-Hg[9], and Pd-Hg[9]. So far, Pd-Hg core-shell nanoparticles represent the most active catalysts in the acidic environment. Benefiting from its optimal HOO* binding energy, core-shell Pd-Hg has been reported to show five times higher mass activity (~130 A g−1) than polycrystalline Pt-Hg/C (~25 A g−1 at 0.65 V vs. reversible hydrogen electrode or RHE, all the potential values are referred to RHE unless specified) with selectivity up to 95% between 0.35 and 0.55 V[9]. However, the high toxicity of Hg might hinder its industrial application. Fe-N-C[26] and Co-N-C[27] are considered as more cost-effective catalysts, but their selectivity needs to be significantly improved.

Herein, we report that direct metal-oxygen coordination can create unique active sites that enable efficient and a more practical electrocatalyst for the 2e− ORR in acidic electrolytes. Specifically, a class of catalysts containing Pd-O-C type coordination can be synthesized by depositing $Pd^{\delta+}$ clusters (3~4 atoms average) onto mildly oxidized CNTs (named as $Pd^{\delta+}$-OCNT in the following context) via a simple solution-impregnation method. Such electrocatalysts show a high $H_2O_2$ selectivity of 95–98% in a wide potential range of 0.3–0.7 V. The onset potential of $Pd^{\delta+}$-OCNT for the 2e− ORR is positively shifted by ~320 mV compared with the OCNT substrate. The mass activity of $Pd^{\delta+}$-OCNT (i.e., 1.946 A mg−1 at 0.45 V) even surpasses that of the core-shell $Pd_2Hg_5$/C[9] by 50%, representing the best reported electrocatalysts for $H_2O_2$ synthesis in acidic electrolytes. Density functional theory (DFT) calculations suggest that the coordination between partially oxidized Pd cluster and OCNT is the key for the enhanced performance of $H_2O_2$ production. Combined with extended X-ray absorption fine structure (EXAFS) characterization, the stable active sites in Pd clusters are identified to be $Pd_3$ and $Pd_4$, with Pd being in the bonding environment of both Pd-Pd and Pd-O. The activity of oxygen-modified $Pd_3$ and $Pd_4$ is further enhanced by a nearby epoxy functional groups, placing the $Pd^{\delta+}$-OCNT catalyst at the peak of the activity volcano with zero overpotential.

## Results

**Understanding the effect of defects on OCNTs for $H_2O_2$ selectivity.** Due to their superior selectivity and activity demonstrated in alkaline electrolytes, we selected OCNTs as a substrate to explore the potential active sites of defect carbons for acidic $H_2O_2$ synthesis. As the ORR overpotential was identified to be too high in acidic electrolytes[19,20], we mainly focused on understanding and optimizing the effect of compositional and structural defects on their 2e− ORR selectivity with the aim to create a functional support that can be used to integrate a second motif that may significantly improve the overall 2e− ORR activity.

CNTs were oxidized under different time durations from 2.5 h to 8.5 h in a nitric acid solution to tune the amounts and types of defects. Transmission electron microscopic (TEM) images (Supplementary Fig. 1) show that the density of defect sites increased with longer oxidation time. When reacted for 6.5 h, abundant defect sites were clearly observed from the changes of the surface roughness and curvature at the OCNTs, suggesting that the bended regions of CNTs were more easily oxidized due to the higher strain than the straight tube walls[28]. After 8.5 h of oxidation, thinner OCNTs with smooth surfaces were observed, which was likely due to the complete etching of the outer-walls of OCNT, a phenomenon also reported by Su et al.[29]. Under all the explored oxidation conditions, tubular nanostructure and crystallinity were maintained as suggested by both TEM and X-ray diffraction (XRD) (Supplementary Fig. 2). Such a defect formation process was also confirmed by the increased intensity ratios of their D and G bands ($I_D/I_G$) in the Raman spectra (Supplementary Fig. 3), changes of surface area (Supplementary Figs. 4 and 5), and the mass loss of the CNTs (Supplementary Fig. 6). Also, the strong signal from the D′ band implied the existence of basal plane $sp^2$ carbon oxidization sites in the OCNT[30].

The oxidation process also introduced defects and functional groups on the surface of OCNTs. Fourier-transform infrared spectroscopy (FTIR) measurements not only confirmed the existence of defects (–$CH_3$) in the samples, but also revealed that the functional groups were mainly C–O and C=O (Supplementary Fig. 7), which were further quantified by X-ray photoelectron spectroscopy (XPS; Supplementary Figs. 8–12). With increased oxidation time from 2.5 to 6.5 h, the percentages of C=C ($sp^2$ carbon) decreased rapidly while the C–O group as the major component of oxygen-containing functional groups increased from 20.1% to 34.6% (on the C basis). Further extending the oxidation time to 8.5 h led to negligible change of C=C groups but a decrease of C–O ratio by ~5%. At the same time, the density of the C–C (structure defects with the form of $sp^3$ carbon) group increased from 4.9% to 11.1%, while that of the C=O groups remained at ~5% during the entire oxidation time.

To correlate the defect characteristics with electrochemical properties, the OCNTs were examined in 0.1 M $HClO_4$ by cyclic

voltammetry (CV), rotating disk electrode (RDE), and rotating ring disk electrode (RRDE). As shown in the CV curves (Supplementary Fig. 13a), the pseudocapacitive current (represented by the redox peaks between 0.2 and 0.8 V) of the OCNT electrodes first increased as the oxidation time extended from 2.5 to 6.5 h and then maintained roughly the same from 6.5 to 8.5 h. The trend of capacitance changes from the redox current was similar with that of the relatively ratios of the C–O groups on the surface from the XPS results (Supplementary Fig. 13b), suggesting that redox peaks could be attributed to the oxidation/reduction of surface quinoidal functional group[31]. All the OCNTs presented a similar onset potential (~0.38 V) to initiate the 2e⁻ ORR. From both the RDE and RRDE tests, OCNTs obtained after 6.5 h oxidation presented the highest $H_2O_2$ selectivity among all the OCNTs, with 95% at 0.1 V through the Koutecky-Levich (K-L) calculation and 90–92% in the range of 0.25 V to 0.35 V from the RRDE measurement (Supplementary Figs. 14–18). It was previously shown that a higher oxygen content in OCNTs resulted in a higher selectivity for $H_2O_2$ in the alkaline medium[13]. In the acidic electrolyte, we found that both the defects and oxygen-containing groups played important roles in determining the 2e⁻ ORR selectivity (Supplementary Fig. 19). For example, when the defect ratio (calculated from deconvoluted C1s XPS peak) increased from 9.7% (6.5 h-OCNT) to 11.1% (8.5 h-OCNT) with a similar number of C–O groups, the $H_2O_2$ selectivity decreased from 90% to 78% at 0.25 V. This result indicated that the defect sites might present strong binding with OH* and O*, leading to more preferred 4e⁻ ORR competing process to produce $H_2O$[32,33].

**Effect of partially oxidized Pd clusters on enhancing 2e⁻ ORR activity.** $Pd^{\delta+}$-OCNT electrocatalysts composed of Pd clusters ($Pd_3$ and $Pd_4$) supported on OCNTs were prepared by loading ~1.0 wt.% of Pd on OCNTs with 6.5 h oxidation, which was identified to be the best substrate (Methods). After Pd deposition, Pd clusters were obtained since no crystalline Pd lattice was detected in the high-resolution TEM (HRTEM) image (Fig. 1a). The Pd clusters were distributed uniformly with a narrow size range of 0.61 ± 0.07 nm on OCNTs (Fig. 1b and Supplementary Fig. 20). However, with mild thermal annealing (450 °C for 5 h) (sample named as H-Pd-OCNT), Pd nanoparticles (3~10 nm) were formed (Fig.1c and Supplementary Fig. 21). XRD patterns clearly show the amorphous nature of the as-deposited Pd in the $Pd^{\delta+}$-OCNT and high crystallinity of Pd in the H-Pd-OCNT sample (Fig. 1d).

The binding environments of $Pd^{\delta+}$-OCNT and H-Pd-OCNT were further characterized using EXAFS (Fig. 1e,f and Supplementary Tables 1–4). The coordination number (CN) of Pd-Pd and Pd-O in $Pd^{\delta+}$-OCNT was found to be 2.5 and 2.7, respectively, suggesting that Pd was coordinated to both Pd and O in the small clusters, and the Pd clusters were partially oxidized. In contrast, the H-Pd-OCNT sample was characterized by a Pd-Pd CN of 7.9, which represented a larger metallic Pd particle (> 3nm) and was consistent with the TEM results.

After deposition of Pd clusters, the surface properties of different samples were further compared. The $I_D/I_G$ ratio in the Raman spectra (Fig. 2a) was 1.82 and 1.71 for $Pd^{\delta+}$-OCNT and H-Pd-OCNT, respectively, showing negligible changes of defects after Pd deposition and heat treatment as compared with the

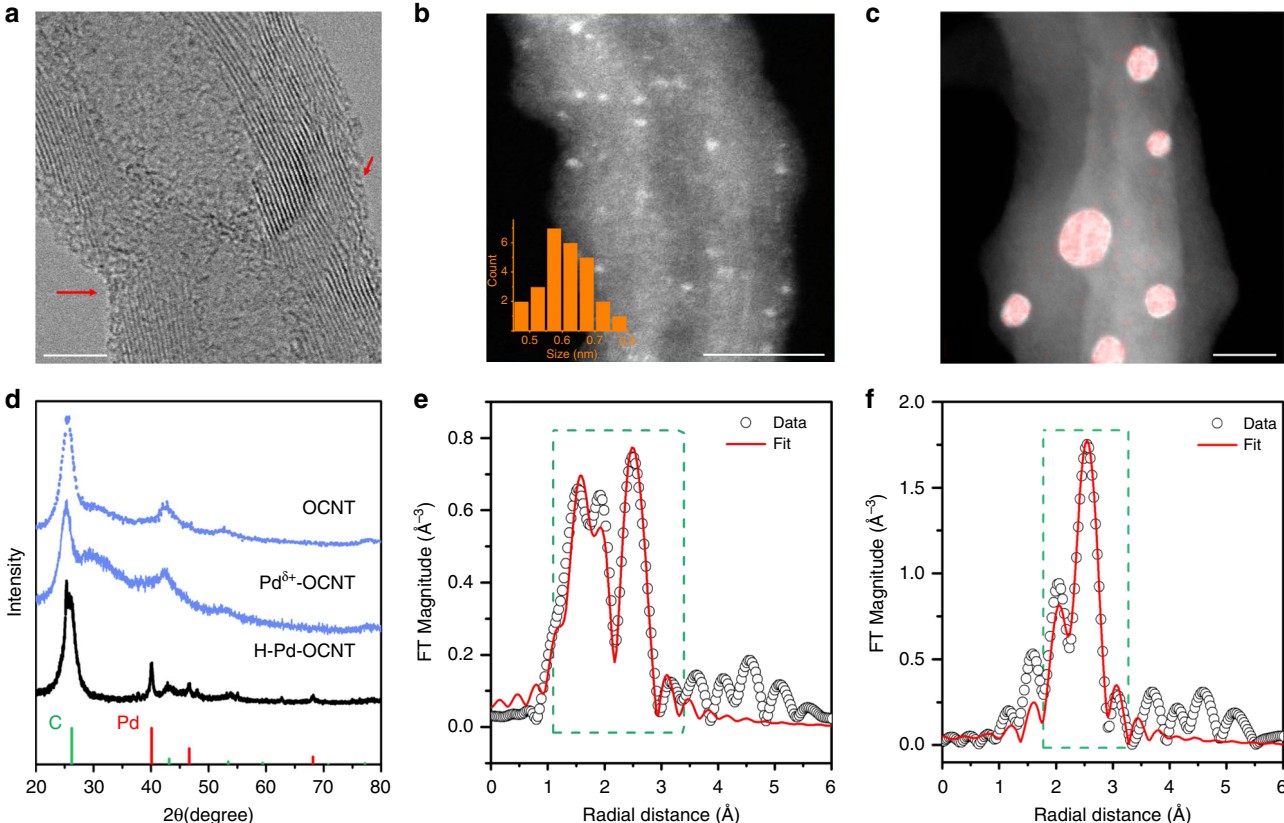

**Fig. 1 Structural characterization of $Pd^{\delta+}$-OCNT and H-Pd-OCNT electrocatalysts. a** HRTEM and **b** annular dark-field (ADF)-STEM image of $Pd^{\delta+}$-OCNT presenting uniform distribution of amorphous Pd atom clusters (scale bar: **a** 5 nm and **b** 10 nm). Inset figure shows the size distribution of the Pd clusters. **c** EDS element mapping of H-Pd-OCNT shows the formation of aggregated crystalline Pd nanoparticles with size of 3-10 nm (red: Pd and scale bar: 10 nm). **d** Powder XRD patterns of 6.5 h OCNT, $Pd^{\delta+}$-OCNT and H-Pd-OCNT. Fourier transform EXAFS analysis of Pd K-edge data for $Pd^{\delta+}$-OCNT (**e**) and H-Pd-OCNT (**f**).

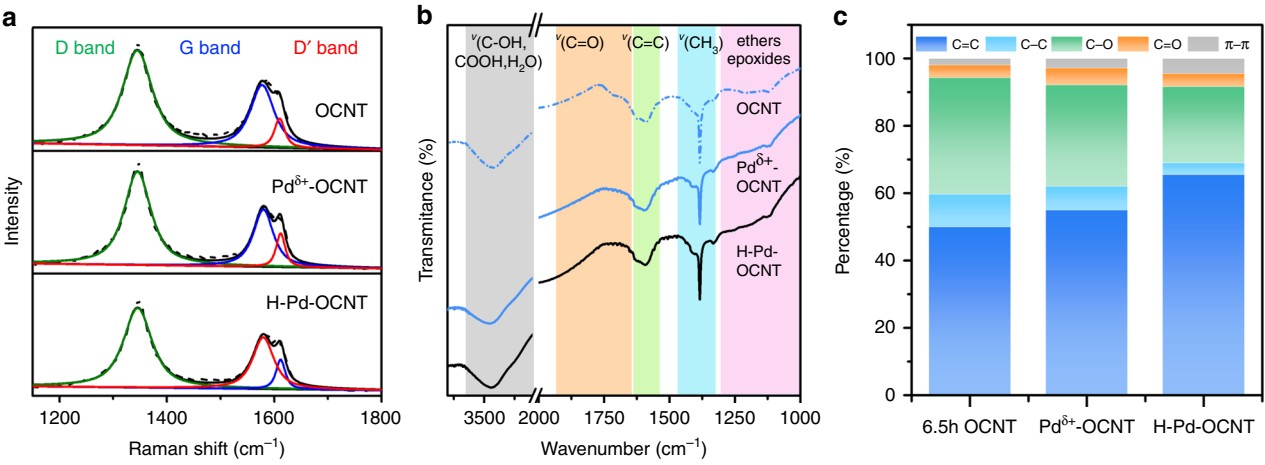

**Fig. 2 Characterization of defects and functional groups in Pd$^{\delta+}$-OCNT and H-Pd-OCNT electrocatalysts.** Raman spectra **a** and FTIR spectra **b** of 6.5 h OCNT, Pd$^{\delta+}$-OCNT and H-Pd-OCNT. Peak assignments are listed in the Supplementary Figure 7**c** The distribution of carbon element in different coordination environments for 6.5 h OCNT, Pd$^{\delta+}$-OCNT and H-Pd-OCNT measured by C1s XPS.

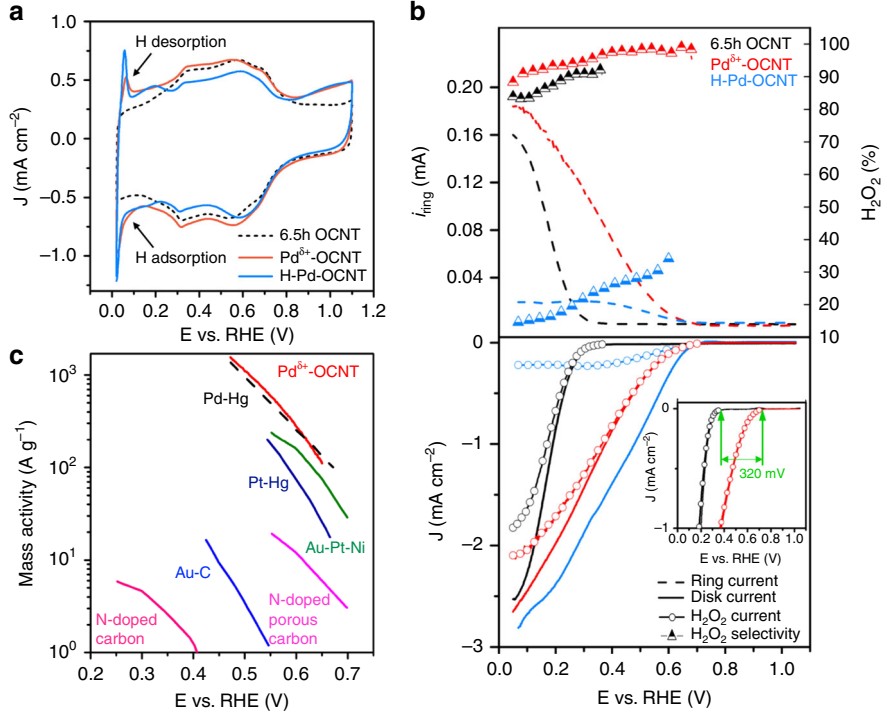

**Fig. 3 Electrochemical performance of Pd$^{\delta+}$-OCNT and H-Pd-OCNT catalysts. a** CV curves of different electrocatalysts showing distinct H adsorption/desorption characteristics at a scan rate of 50 mV s$^{-1}$. The electrolyte was Ar-saturated 0.1 M HClO$_4$ solution. **b** RRDE voltammograms in O$_2$-saturated HClO$_4$ electrolyte with a scan rate of 10 mV s$^{-1}$ at 1600 rpm (only the anodic cycle is shown). The H$_2$O$_2$ current and selectivity were calculated from the ring and disc currents for both OCNT and Pd$^{\delta+}$-OCNT. **c** Mass activity of the state-of-the-art electrocatalysts for H$_2$O$_2$ production in the acid electrolyte. Data were taken from previous reports[8, 9, 15, 20, 24, 35, 36].

OCNT ($I_D/I_G = 1.88$). Also, the basal plane $sp^2$ carbon oxidization sites still remained in both Pd$^{\delta+}$-OCNT and H-Pd-OCNT as shown by the $D'$ band. FTIR results indicated that the types of surface functional groups (C=O, C–O) were maintained after Pd deposition and annealing (Fig. 2b). XPS results also showed similar abundancy of $sp^3$ carbon defects, C–O and C=O with OCNTs, further suggesting that the deposition of Pd clusters did not change the surface properties of the OCNTs (Fig. 2c, Supplementary Fig. 23). For H-Pd-OCNT, the ratio of $sp^3$ carbon defects and C–O group decreased with an increase of C=C ratio (Supplementary Fig. 24), likely due to the cleavage of less

thermally stable functional groups under annealing[34]. Such a catalyst structure allows us to further identify the unique role of the partially oxidized Pd clusters in catalyzing the 2e$^-$ ORR by isolating the effect of Pd and the defect carbon substrates.

The effect of Pd clusters on the H$_2$O$_2$ selectivity and activity was investigated by comparing Pd$^{\delta+}$-OCNT with OCNT and H-Pd-OCNT. The characteristic hydrogen adsorption/desorption peaks of Pd in both Pd$^{\delta+}$-OCNT and H-Pd-OCNT electrocatalysts were observed in the CV curves (Fig. 3a), confirming the successful loading of Pd onto the OCNT surface. The most interesting feature for the Pd$^{\delta+}$-OCNT catalyst was observed in

the ORR process (Fig. 3b). After the introduction of Pd clusters, both the ring and disc currents of $Pd^{\delta+}$-OCNT initiated earlier than that of OCNT, resulting in a positive shift of ORR onset potential by ~320 mV. The $H_2O_2$ current of the $Pd^{\delta+}$-OCNT electrocatalyst nearly overlapped with total reaction (disc) current, which suggested that the ORR almost exclusively proceeded toward the $2e^-$ pathway. The calculated $H_2O_2$ selectivity of the $Pd^{\delta+}$-OCNT catalyst was in the range of 98% to 95% in the potential range of 0.7 to 0.3 V (Fig. 3b and Supplementary Fig. 26), superior to precious metal-based electrocatalysts reported previously[8,9,22,23,25]. More importantly, the kinetic mass activity of $Pd^{\delta+}$-OCNT for $H_2O_2$ production reached 1.946 A mg$^{-1}$ at 0.45 V, about 1.5 times of the core-shell $Pd_2Hg_5$/C catalyst[9] and significantly higher than that of other electrocatalysts in acidic electrolytes (Fig. 3c)[8,9,15,20,24,35,36]. As for the H-Pd-OCNT catalyst, although it showed a similar positive shift in onset potential as $Pd^{\delta+}$-OCNT, it preferred the $4e^-$ ORR pathway to completely reduce $O_2$ to $H_2O$, showing only 18% of $H_2O_2$ selectivity at 0.1 V. Thus, we conclude that the partially oxidized Pd clusters are the key in enhancing activity and maintaining high selectivity for $H_2O_2$ production.

To demonstrate their viability for continuous ORR in fuel cell operations, we deposited the $Pd^{\delta+}$-OCNT electrocatalysts on carbon paper as a working electrode and fabricated a device that could synthesize $H_2O_2$ directly in acidic electrolyte (Supplementary Fig. 28). In such a device, $O_2$ was reduced to yield $H_2O_2$ directly by combining with the protons in the acidic electrolyte without the need of molecular $H_2$. The amount of $H_2O_2$ generated in an H-cell was obtained by a titration method. When the catalyst mass loading was controlled to 0.1 mg cm$^{-2}$, a steady current density of 10 mA cm$^{-2}$ was recorded at 0.1 V (Fig. 4a). The selectivity of $H_2O_2$ was measured to be 87%, which was close to the RRDE test at 0.1 V. Also, the yield of $H_2O_2$ was

up to 1701 molkg$_{cat}$$^{-1}$ h$^{-1}$, 2 times higher than that of the single atomic Pt electrocatalyst reported recently[37]. Most importantly, the end $H_2O_2$ concentration reached 10 wt% after the durability test, which could be readily used for acid-based chemical synthesis (9 wt% is commonly used)[6]. The ORR stability of $Pd^{\delta+}$-OCNT was evaluated by chronoamperometry (CA) test by holding the disk electrode potential at 0.1 V for more than 8 hr. Both the disc and ring currents decreased by only ~15% after the test and the $H_2O_2$ selectivity was still maintained at 86% (Fig. 4b). It was found that the morphology and size distribution of Pd clusters on OCNTs showed negligible changes after the stability test (Fig. 4c,d), which was responsible for their good electro-chemical stability during the ORR.

The enhanced $2e^-$ ORR performance of $Pd^{\delta+}$-OCNT was further investigated by DFT calculations. Since the diameter of CNT in the experiment was 10 to 20 nm, a negligible strain energy is expected hence a two-dimensional graphene sheet was used as a model structure[38]. The Pd clusters in defect CNT were first studied by modeling a variety of Pd clusters ranging from 1 to 4 Pd atoms trapped in the vacancies of the graphene substrate (Supplementary Fig. 31a). For $Pd_1$, the possibility of a Pd atom being trapped in either single vacancy or double vacancy of graphene was considered. Larger vacancies were required to trap the $Pd_2$, $Pd_3$ and $Pd_4$ clusters. For $Pd_2$ and $Pd_3$ a vacancy with at least 3 missing carbon atoms was required, while for $Pd_4$ a vacancy with 4 missing carbon atoms was a prerequisite to form a sufficiently stable structure. The $2e^-$ ORR proceeds via $1e^-$ reduction of $O_2$ to HOO* and subsequent $1e^-$ reduction of HOO* to $H_2O_2$ where both reduction steps involve HOO* as the sole intermediate. It has been shown that the adsorption energy of HOO* was the key activity descriptor for the $2e^-$ ORR, where the maximum activity observed at an optimized binding of the HOO* intermediate[8,9]. Therefore, the HOO* adsorption energies

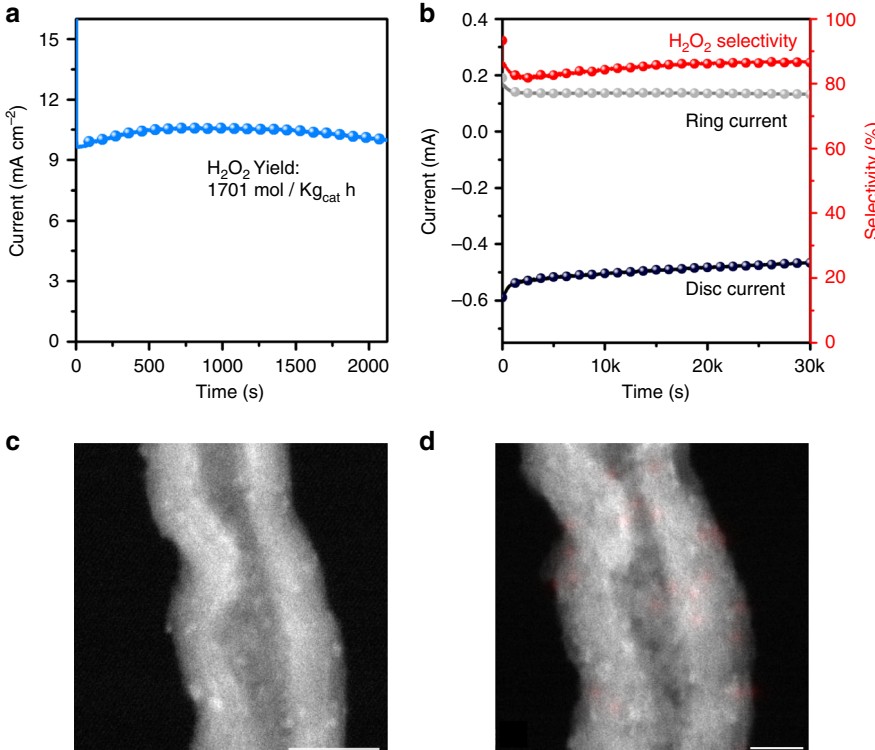

**Fig. 4 Yield and stability of $H_2O_2$ electrocatalysts in the acid electrolyte. a** Chronoamperometry curve of $Pd^{\delta+}$-OCNT in the H-cell test at 0.1 V. **b** Stability test of $Pd^{\delta+}$-OCNT in a $O_2$-saturated 0.1 M $HClO_4$ at 0.1 V. All the experiments were performed at 25 °C. ADF-STEM image **c** and its corresponding EDS element mapping **d** of $Pd^{\delta+}$-OCNT after the stability test (scale bar: **c** 10 nm and **d** 5 nm).

were calculated on all the model structures. The results are summarized in Supplementary Fig. 31b in the form of the free energy diagram at the equilibrium potential of the 2e⁻ ORR (0.70 V). An ideal catalyst should have a flat free energy diagram at this potential (0.70 V), exhibiting highest catalytic activity with zero overpotential. This plot shows that none of the examined structures are sufficiently active for 2e⁻ ORR as they all bind HOO* too strongly such that further reduction of HOO* to $H_2O_2$ becomes a bottleneck. Consequently, the bare Pd clusters trapped in the graphene vacancies are not likely the active sites for the 2e⁻ ORR. In fact, the strong tendency of Pd clusters toward adsorbing HOO* results in dissociating the HOO* species to form HO* and O*, indicating that metallic Pd atoms prefer 4e⁻ ORR. Next, the effect of oxidation, both in the Pd clusters and CNT, was investigated to unravel the active sites responsible for the high 2e⁻ORR activity observed in the experiment.

At the potential of 0.70 V it is highly likely that the Pd clusters are covered with several O*, HO* species or a combination of both. To study the oxygenated species coverage effect, only $Pd_3$ and $Pd_4$ were considered, which were consistent with the experimental measurements of the Pd cluster size (0.6 nm). Figure 5a and Supplementary Figure 32 display the calculated formation energy of a variety of possible O*/HO* coverages on $Pd_3$ cluster as a function of applied potential. The results for the $Pd_4$ cluster are reported in the Supplementary Figs. 33 and 34. The lowest line at 0.70 V displays the most stable coverage in each case. For $Pd_3$ and $Pd_4$, 3 O*/HO* and 3HO* were the steady state oxygen coverage, respectively. This analysis suggests the presence

of the Pd-O bonds, which is in agreement with the EXAFS results. We further took these partially oxidized $Pd_n$ clusters and calculated the HOO* adsorption energy to model the 2e⁻ ORR and to identify trends in activity. The results are summarized in Fig. 5b in the free energy diagram plot indicating that the oxygen coverage on the $Pd_n$ clusters improves the HOO* adsorption energy and brings it closer to the range with high ORR activity.

## Discussion

The CNT substrate was already oxidized from the experimental results, further DFT calculations were performed to examine the effect of neighboring oxygen functional groups on the HOO* adsorption energy. As an example, an oxygen-containing functional group such as epoxy was used to account for the C–O moiety. Figure 5c displays the atomic structures in the presence of two nearby epoxy groups. The results are summarized in an activity volcano plot (Fig. 5d). The calculated limiting potential ($U_L$) is used as an indicator of activity toward the 2e⁻ ORR, which is defined as the maximum potential at which both 1e⁻ reduction of $O_2$ to HOO* and subsequent 1e⁻ reduction of HOO* to $H_2O_2$ are downhill in free energy. The maximum activity is therefore achieved at the HOO* binding energy of 4.22 eV, which corresponds to the value at the peak of volcano. The results show that the presence of a nearby functional group further improves the activity of both $Pd_3$ and $Pd_4$ and places them at the peak of the activity volcano with zero overpotential. It also shows that the synergy between the oxygen coverage and oxygen functional group plays an important role in improving the

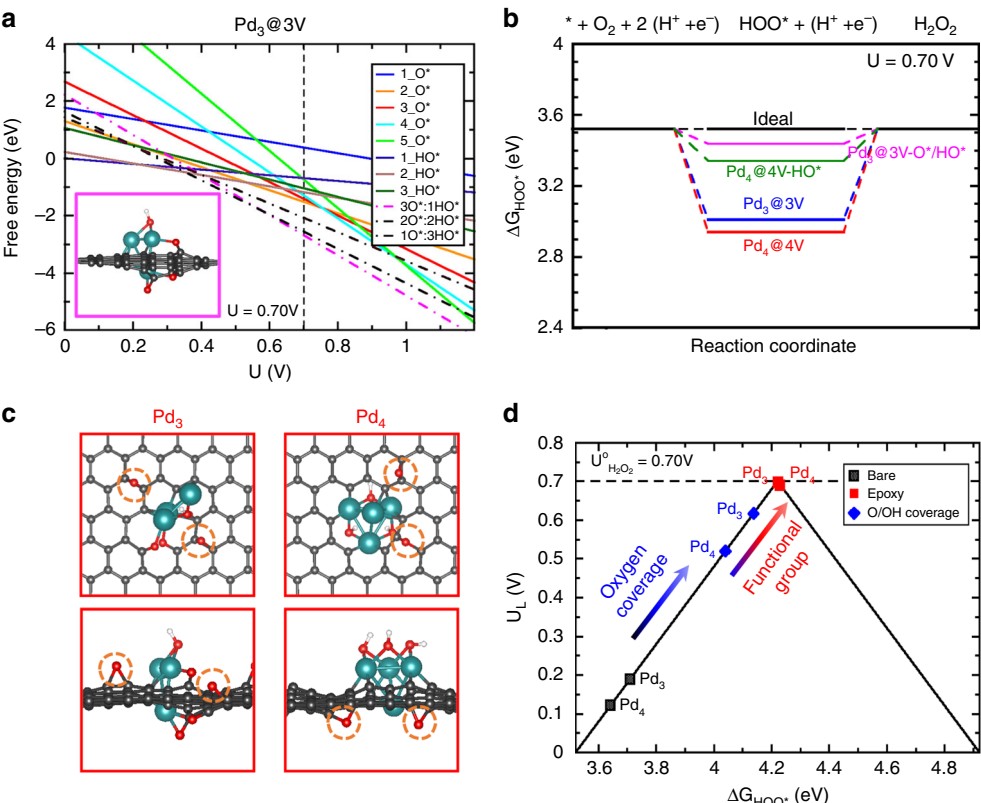

**Fig. 5 DFT calculations of ORR activity and selectivity on different motifs. a** Pourbiax diagram for determining the steady state coverage of the oxygenated species on $Pd_3$ under 2e⁻ ORR standard redox potential (0.70 V). The inset shows the side view of the most stable coverage. Color code for the atomic structure; C: gray, O: red, Cyan: Pd. **b** Free energy diagram for 2e⁻ ORR over the most stable structure from Pourbiax analysis at 0.70 V. **c** Optimized DFT model structures with nearby epoxy functional groups (highlighted by orange dashed circles) on the most stable O*/HO* covered $Pd_3$ and $Pd_4$ clusters. **d** Activity volcano plot, the y axis is the calculated limiting potential ($U_L$), defined as the maximum potential at which the reaction steps become downhill in free energy. The x axis is the calculated free energy of HOO*. Horizontal dashed line is the standard redox potential for the 2e⁻ ORR (0.70 V).

catalytic activity of small Pd clusters anchored in OCNT. Therefore, we conclude that the high activity and selectivity observed in the experiments is a direct consequence of the synergy between partially oxidized Pd clusters and oxidized CNT substrate. These two effects together significantly improve the $2e^-$ ORR activity while maintaining high selectivity.

In summary, we demonstrated a class of $2e^-$ ORR electrocatalysts by the synergistic interaction between partially oxidized Pd clusters and oxygen-functionalized CNT substrate. Through a simple solution-impregnation method, $Pd_3$ and $Pd_4$ clusters can be readily deposited on OCNTs with the CN of Pd-Pd and Pd-O of 2.5 and 2.7, respectively, as confirmed by the EXAFS characterization. The unique $Pd^{\delta+}$-OCNT electrocatalyst showed high $H_2O_2$ selectivity at 95–98% in a wide potential range of 0.3 to 0.7 V and a positive shift of the $2e^-$ ORR onset potential by ~320 mV compared with the OCNT substrate. The mass activity of $Pd^{\delta+}$-OCNT was 1.946 A mg$^{-1}$ at 0.45 V, 1.5-fold higher than $Pd_2Hg_5$/C, which was the best electrocatalyst reported for $H_2O_2$ synthesis in acidic electrolytes. In addition, the $H_2O_2$ yield rate was estimated to be 1700 mol kg$_{cat}^{-1}$ h$^{-1}$ in an H-cell test and the $Pd^{\delta+}$-OCNT electrocatalyst maintained excellent stability with no decrease of the $H_2O_2$ selectivity above 8 h of testing, suggesting its promise for the electrochemical synthesis of $H_2O_2$. DFT calculations further suggest that the coordination between oxygen-modified Pd clusters and the oxygen-containing functional groups on OCNT is the key for their high selectivity and activity for $2e^-$ ORR. This work offers a unique path toward the development of highly selective ORR electrocatalysts by simply tuning the interactions between the active metal and the oxidized carbon support.

## Methods

**Preparation of OCNTs and $Pd^{\delta+}$-OCNT**. To prepare O-CNTs with different density of defects and functional groups, 250 mg of multi-walled CNTs (produced by a tons-scale fluidized chemical vapor deposition process)[39] was refluxed in 20 ml of HNO$_3$ (Fisher Scientific, 68 wt. %) for 2.5 h, 4.5 h, 6.5 h, or 8.5 h at 140 °C. The resulting product was obtained after centrifugal separation and drying at 55 °C. A simple impregnation method was used to prepare Pd supported by OCNTs ($Pd^{\delta+}$-OCNT). Specifically, 2.5 mg of PdCl$_2$ (Alfa Aesar, 99.9%) and 50 mg of OCNT were suspended in 20 ml of 7 wt.% HNO$_3$ solution and heated at 65 °C with vigorous stirring until the mixture was fully dried. To anneal the $Pd^{\delta+}$-OCNT, the as-prepared sample was heated from room temperature to 100 °C at a rate of 10 °C min$^{-1}$ and kept at 100 °C for 1 h under argon (Ar) protection before ramping to 450 °C at a rate of 4 °C min$^{-1}$. Then it was annealed for 5 h at this temperature to obtain thermally annealed sample (H-Pd-OCNT).

**Characterization**. The defect formation process and distribution Pd clusters of different samples were characterized by high-angle annular dark-field scanning TEM (Hitachi HD 2700C). Energy dispersive X-ray spectroscopy was performed by FEI Talos F200X to obtain element distributions of Pd on each sample. The structure and phase composition were further characterized by X-ray diffractometer (XRD, Bruker AXS) equipped with a Cu Kα radiation source ($\lambda$ = 1.5406 Å). The specific mass loading of the Pd atomic clusters was determined by inductively coupled plasma mass spectrometry (iCAP Qc, Thermo Fisher Scientific). To investigate the heteroatoms and functional groups, a commercial SPECS Ambient-pressure X-ray photoelectron spectrum chamber combined with a PHOIBOS 150EP MCD-9 analyzer and FTIR (Nicolet iS50) were used. The Raman spectra were acquired by a Renishaw inVia with 532 nm laser source. Nitrogen adsorption/desorption were conducted by an autosorb iQ2.

**Electrochemical measurements**. Electrochemical test was performed in three-electrode cells, where a graphite and Ag/AgCl (3 M Cl$^-$) were used as the counter electrode and reference electrode, respectively. The electrocatalyst inks were prepared by dispersing samples in a Milli-Q and isopropanol solution (4 : 1) with 10 μl of Nafion (5%) to achieve the mass concentration of 1 mg ml$^{-1}$ for $Pd^{\delta+}$-OCNT and H-Pd-OCNT samples, and 3.5 mg ml$^{-1}$ for O-CNT samples. Ten microliters of each catalyst ink was then deposited on a pre-cleaned glassy carbon electrode (0.196 cm$^{-2}$). The CV curves were recorded in Ar-saturated 0.1 M HClO$_4$ electrolyte with a scanning rate of 50 mV s$^{-1}$. The ORR performance was examined by RDE and RRDE in an O$_2$-saturated 0.1 M HClO$_4$ solution at a scanning rate of

10 mV s$^{-1}$ with capacity current correction (in Ar-saturated 0.1 M HClO$_4$). The ring current was hold at 1.2 V (vs. RHE) to further oxidize the as-formed $H_2O_2$ and collection efficiency was calibrated to be 0.37. The stability test was performed by CA test at 0.1 V for 30,000 s. The selectivity was calculated by previous report[6,10], detailed below.

The $H_2O_2$ selectivity of samples based on RDE was calculated by K-L plot in Eqs. (1, 2) from the polarization curves at different rotation speeds.

$$\frac{1}{j} = \frac{1}{j_{kin}} + \frac{1}{j_{diff}} = \frac{1}{j_{kin}} + \frac{1}{B \cdot \sqrt{\omega}} \tag{1}$$

$$B = 0.62 \cdot n \cdot F \cdot D_{O_2}^{2/3} \cdot \nu^{-1/6} \cdot C_{O_2} \tag{2}$$

where $j$ is the current density consists of a kinetic current ($j_{kin}$) and a diffusion current ($j_{diff}$), $\omega$ is the rotation speed, $n$ is the number of electrons transferred during the reaction, and $D_{O_2}$ and $C_{O_2}$ are the diffusivity and solubility of oxygen, respectively; $F$ is the Faraday constant and $\nu$ is the kinematic viscosity of the electrolyte. For a $4e^-$ process, $B = 0.47$ mA cm$^{-2}$ s$^{1/2}$ and for a $2e^-$ process, $B = 0.23$ mA cm$^{-2}$ s$^{1/2}$ [19]. For RRDE tests, the $H_2O_2$ selectivity was calculated by Eq. (3).

$$H_2O_2(\%) = 200* \frac{I_R/N}{I_D + I_R/N} \tag{3}$$

where $I_R$ and $I_D$ are the ring current and disk current, respectively, and $N$ is the collection efficiency.

To further confirm the selectivity of the $Pd^{\delta+}$-OCNT electrocatalyst, a H-cell with a Nafion 117 membrane was used. Electrocatalysts were loaded on Telfon-treated carbon papers (0.1 mg cm$^{-2}$). The concentration of generated $H_2O_2$ was measured by its reaction with Ce(SO$_4$)$_2$ (2Ce$^{4+}$ + H$_2$O$_2$ → 2Ce$^{3+}$ + 2 H$^+$ + O$_2$). The color of solution changes from yellow to colorless through the reaction. The concentration of Ce$^{4+}$ after the reaction was measured by ultraviolet–visible spectroscopy (Perkin Elmer UV-VIS-NIR Spectrometer) with 316 nm of wavelength.

**X-ray absorption fine structure measurements**. XAFS measurements were conducted in the 7-BM beamline (QAS) at National Synchrotron Light Source-II (NSLS-II) at Brookhaven National Laboratory. Both transmission and fluorescent signals were detected. The typical duration for a single spectrum was 47 sec and thirty spectra were merged to get high signal-to-noise spectrum. During all of the XAFS measurements, the spectrum of reference Pd foil was simultaneously recorded, and was further used for calibrating the edge energy ($E_0$) of the sample under analysis.

The obtained spectra were processed using the ATHENA and ARTEMIS software in IFFEFIT package[40–42]. The procedure which was described in Ravel et al.[41], was followed during the data process[40]. EXAFS analyses were conducted by using the ARTEMIS software. The EXAFS spectrum ($\chi(k)$) was weighted with $k^2$ value to intensify the signal at high $k$-regime. The Hanning window was utilized for the Fourier transform. All of the EXAFS fitting was done in the $R$-space. The goodness of fitting was evaluated based on the reliable factor (R-factor) and reduced chi-square (reduced $\chi^2$). The fitting results are tabulated in Supplementary Tables 1–4.

**Computational methods**. Atomic Simulation Environment[43] was used to handle the simulation and the QUANTUM ESPRESSO[44] program package to perform electronic structure calculations. The electronic wavefunctions were expanded in plane waves up to a cutoff energy of 500 eV, while the electron density is represented on a grid with an energy cutoff of 5000 eV. Supplementary Fig. S35 displays the energy cutoff convergency plots for calculated adsorption energies and limiting potentials for an example of our model calculations ($Pd_2$ cluster). Core electrons were approximated using ultrasoft pseudopotentials[45]. To describe chemisorption properties on graphene structures, PBE exchange-correlation functional with dispersion correction was used[46]. Graphene structures were modeled as one layer with a vacuum of 20 Å to decouple the periodic replicas. A $5 \times 5$ super cell lateral size was used to model $Pd_1$ and $Pd_2$ clusters and the Brillouin zone was sampled with ($4 \times 4 \times 1$) Monkhorst–Pack $k$-points. For the larger clusters of $Pd_3$ and $Pd_4$ we used a $7 \times 7$ super cell lateral size with ($2 \times 2 \times 1$) Monkhorst–Pack $k$-points sampling. Supplementary Fig. S36 displays the $k$-point sampling convergency plots for the calculated adsorption energies and limiting potentials for an example of our model calculations ($Pd_3$ cluster covered with 3 O* and 1HO*).

## Data availability

The source data underlying Figs. 1–4 are provided in the Supplementary Information. The data that support other plots within this paper are available from the corresponding author upon reasonable request.

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

## Acknowledgements

Z.C. acknowledges the support from the ACS Petroleum Research Fund (#59989-DNI5) and Jacob School of Engineering at UC San Diego. The XPS measurements and RRDE experiments performed at Columbia University were supported by the U.S. Department of Energy, Office of Science, Catalysis Science Program (DE-FG02-13ER16381). Electron microscopy work was performed at the Center for Functional Nanomaterials, BNL, which is supported by the U.S. Department of Energy (DOE), Office of Basic Energy Science, under contract DE-SC0012704. We acknowledge technical supports with 7-BM (QAS) at National Synchrotron Light Source-II (NSLS-II) and Center for Functional Nanomaterials (CFN) in Brookhaven National Laboratory (Contract Number DE-SC0012704). S.S. and A.H.B.M. acknowledge the support from the University of Calgary's Canada First Research Excellence Fund Program, the Global Research Initiative in Sustainable Low Carbon Unconventional Resources.

## Author contributions

Z.C., J.G.C., and Q.W.C. designed the experiment. Q.W.C. carried out the electrochemical evaluations and analyzed the data. P.Z. synthesized samples. A.H.B.M. and S.S. conducted DFT calculations. X.R.Z. and H.L.X. conducted the TEM experiments. S.R.D. conducted the XPS test and analyzed the data. J.H.L. conducted the synchrotron-based experiment and analyzed the data. H.P.G. conducted the Raman test and Y.Z. conducted the BET measurement. Z.C. and J.G.C. supervised the whole project.

## Competing interests

The authors declare the following competing interests: a patent was filed for this work through the UCSD Office of Innovation and Commercialization.
