## [Peer Review File · Nature Communications]

Reviewers' Comments:

Reviewer #1:

Remarks to the Author:

The manuscript by Chang et al reports a study on the selective production of H₂O₂ via ORR on Pt clusters on O-containing C nanotubes. Even if the system is relatively new, the conclusions are not particularly new: it is well known that ORR limiting step is the *OOH formation. Moreover, I have some concerns regarding the computational model used; maybe 500 eV is enough as plane wave cutoff because they are using ultra soft pseudo potentials but they are not very accurate. Since DFT has problems when describing the oxygen molecule, the combination of DFT+ultrasoft pseudo potentials is not very reliable when studying oxygen electrocatalysis. Regarding the supercell, a 5x5 supercell of graphene may be large enough for small Pt clusters when only one or two vacancies are considered but for bigger clusters is clearly not sufficient since images do interact and energetics are then compromised by this effect and not comparable with smaller non-interacting clusters. Thus, the volcano plot is not totally reliable. A 2x2 in-plane sampling is really really low ... more than 20x20 MP divisions are needed to get the correct electronic structure of graphene using the unit cell, so at least 4x4 k-point sampling would be needed for a 5x5 supercell. For such lack of novelty and flaws in the computational methodology, I cannot recommend publication of this manuscript in its present form

Reviewer #2:

Remarks to the Author:

In this manuscript, the authors describe the preparation of partially oxidized Pd anchored on defect carbon. In recent years, the ORR performance of materials' in acidic conditions has been widely studied and the authors reported the performance of partially oxidized Pd with oxygen-functionalized carbon in promoting H₂O₂ production via 2 e⁻ oxygen reduction in acidic electrolyte. They illustrated the synergistic effect of Pd and carbon and the materials showed nearly 100% selectivity toward H₂O₂ in a wide potential range of 0.3-0.7 V. This study is largely well carried out, and it is suitable to be published in Nat. Commun. However, some problems should be addressed before this work can be accepted for publication.

(1) In the abstract, the authors described the materials as partially oxidized palladium, however, there were no sufficient experimental evidences to prove that the palladiums were partially oxidized. Therefore, the authors should provide more evidences. And the materials were denoted as Pd-OCNT or H-Pd-OCNT. It maybe improper because the Pd atoms were partially oxidized.

(2) In page 5, the authors stated that the surfaces become smooth after 8.5 h of oxidation due to the complete etching of outer-walls of OCNT. Why? Whether the authors studied the outer-surfaces' roughness when the oxidation time was over 8.5 h and whether the surfaces would become rough or not.

(3) In page 5, the authors stated that the increase of pseudocapacitive current as the oxidation time extended from 2.5 h to 8.5 h could be attributed to the increased number of C-O groups. However, in the XPS results, as stated in the 144 line in page 5, the ratio of C-

O groups was decreased by ~5% when the oxidation time was 8.5 h. So the explanation is not consistent with the XPS results. The authors should think over the results carefully. (4) The captions of supplementary Fig 4 and 24. are not appropriate. For example, the caption of supplementary Fig 4. should be changed as follows: The N₂ adsorption-desorption isotherms of OCNT samples.

Reviewer #3:

Remarks to the Author:

The authors present an interest report into the partially oxidised Pd on CNTs for selective production of H₂O₂ via oxygen reduction in acidic media. The motivations for the work are clear, and the case for Pd-O-C as an efficient catalyst unit is strong and supported by experimental and computational data. The clear distinction between the Pd cluster and larger Pd particles is well defined and supported. In particular, it is encouraging to see the catalyst performing well both under RRDE and H-cell operating conditions. I would like to raise a couple of points that I think will strengthen the paper, but beyond these I am happy to recommend this article for publication.

Line 89: The authors introduce the phrase "optimum *OOH binding energy" before any discussion of ORR mechanism, intermediates or DFT. It may be worth replacing this with a more general description to aid the more general interest reader

Line 72: The authors may wish to include B- or N-doped carbons in their discussions of defective carbon materials for H₂O₂ from ORR

Figures throughout: Unit convention for potential axes should keep V as the unit alone, i.e. E vs RHE (V) rather than E (V vs RHE)

Figure 3B: The current density for H-Pd-OCNT is significantly lower than for Pd-OCNT, though the authors state that the ORR is switching from the 2e⁻ to the 4e⁻ mechanism. This seems counter-intuitive, as moving to the 4e⁻ mechanism should cause a larger magnitude current to pass. Do the authors have an explanation for this hindered current.

Supplementary Figure 18: The triangle and circle symbols in the legend appear to be the wrong way around

Supplementary Figure 19: If the bars stay unattached to an axis, it would be useful to add data labels, or some other way of indicating the magnitude of the selectivity change with reaction time

Supplementary Figure 26: Legend is missing

Line 246: It would be useful to discuss the operation of the H-cell in terms of current

density and to give an idea of the H-cell volume and end concentration of H₂O₂ to give a clear picture of peroxide production rate. Similar scaling information would be useful for supplementary figure 27 as well.

Line 247: The titration method appears to be missing from the supplementary information. Related to this section, does the titration method take H₂O₂ decomposition into account during the H-cell operation?

Similarly, the selectivity of 87% at 5 mA is promising for higher production rates of H₂O₂. Do the authors have any indication of how this material would perform under higher operating conditions $\sim 100 \text{ mA cm}^{-2}$? Is this material a candidate for replacing the anthraquinone process? Specifically, I wonder how stable the partially oxidised environment will be under more harshly reducing conditions.

Figure 5: The interaction with the epoxy unit gives a clear shift towards the top of the volcano. Can the authors comment on how susceptible ΔG_{HOO^*} is to the nature of this surface bound oxygen? Presumably the OCNTs will show a range of different surface bound O?

Reviewer #1 (Remarks to the Author):

(1) The manuscript by Chang et al reports a study on the selective production of H₂O₂ via ORR on Pt clusters on O-containing C nanotubes. Even if the system is relatively new, the conclusions are not particularly new: it is well known that ORR limiting step is the *OOH formation.

Response:

We think there is a misunderstanding here as reviewer indicated Pt clusters in the comment. We would like to emphasize that our study was focused on Pd clusters and the *OOH formation is indeed a well-known rate limiting step for two-electron ORR (Siahrostami, et al., *Nat. Mat.* 2013, 12(12),1137.)¹ and that is exactly why we have investigated this step in our study to make a comprehensive comparison for our system with the previous reports on this topic.

(2) Moreover, I have some concerns regarding the computational model used; maybe 500 eV is enough as plane wave cutoff because they are using ultra soft pseudo potentials but they are not very accurate. Since DFT has problems when describing the oxygen molecule, the combination of DFT+ultrasoft pseudo potentials is not very reliable when studying oxygen electrocatalysis.

Response:

We thank the reviewer for this comment. To address the reviewer's concern, we have shown the adsorption energy of OOH* for 450, 500, 550 and 600 eV cutoff energies on Pd₂ cluster as an example. The results shown in **Figure R1** indicate that higher cutoff energies such as 550 and 600 eV have essentially no impact on the calculated adsorption energy (ΔG_{OOH^*}) (Figure a) and subsequent limiting potential (U_L) (Figure b). We therefore conclude that 500 eV cutoff energy has a reasonable accuracy in predicting both calculated adsorption energies and limiting potentials for our calculations.

Figure R1. Top figure shows the Pd₂ cluster embedded at graphene vacancy in a 5×5 unit cell. a) and b) are the calculated adsorption energy of OOH* and calculated limiting potential for different cutoff energies, respectively.

Action:

To address the reviewer’s comment, we included these figures in the Supplementary Figure 35 in the revised supplementary information.

[Supplementary Figure.35 top figure shows the Pd₂ cluster embedded at graphene vacancy in a 5×5 unit cell. a) and b) are the cutoff energy convergency plots for the calculated adsorption energy of OOH* (our activity descriptor) and calculated limiting potential, respectively.]

(3) Regarding the supercell, a 5x5 supercell of graphene may be large enough for small Pt clusters when only one or two vacancies are considered but for bigger clusters is clearly not sufficient since images do interact and energetics are then compromised by this effect and not comparable with smaller non-interacting clusters. Thus, the vulcano plot is not totally reliable.

Response:

We thank the reviewer for noticing this point and apologize for the mistake we made in not mentioning the correct details in the computational method in the SI. Indeed, our larger Pd

cluster calculations i.e., Pd₃ and Pd₄ have been done in a 7×7 unit cell with 2×2×1 in-plane k-point sampling. For Pd₁ and Pd₂ calculations, we used 5×5 unit cell size with 4×4×1 in-plane k-point sampling. Below (**Figure R2**) are the final optimized structures for the most stable configurations of Pd₃ and Pd₄ shown previously in the Pourbiax diagram (Figure 5a) in the main text and Supplementary Figure 31, including the unit cell size for review. To address the reviewer's point, we clearly explained the details of our calculations and corrected the typos in the revised version of the SI and included the figures below.

Figure R2. Top and side views of our original optimized structures for the Pd₃ and Pd₄ clusters which were done in a 7×7 unit cell to avoid the interaction between periodic images. The side views show the angle we chose for the Pourbiax diagrams in the main manuscript (original version).

Action:

To address the reviewer's comment, we included these figures in the Supplementary Figure 37 and corrected the typos in the methods.

[Supplementary Figure 37. Top and side views of the optimized structures for Pd₃ and Pd₄ clusters which were done in a 7x7 unit cell to avoid the interaction between periodic images. The side views show the angle we chose for the Pourbiax diagrams in the main manuscript and Supplementary Figure 33.]

[Supplementary Figure 35 displays the energy cutoff convergency plots for calculated adsorption energies and limiting potentials for an example of our model calculations (Pd₂ cluster).]

[A 5 × 5 super cell lateral size was used to model Pd₁ and Pd₂ clusters and the Brillouin zone was sampled with (4 × 4 × 1) Monkhorst-Pack k-points. For the larger clusters of Pd₃ and Pd₄ we used a 7 × 7 super cell lateral size with (2 × 2 × 1) Monkhorst-Pack k-points sampling. Supplementary Figure 36 displays the k-point sampling convergency plots for the calculated adsorption energies and limiting potentials for an example of our model calculations (Pd₃ cluster covered with 3O* and 1HO*).]

(4) A 2x2 in-plane sampling is really really low ... more than 20x20 MP divisions are needed to get the correct electronic structure of graphene using the unit cell, so at least 4x4 k-point sampling would be needed for a 5x5 supercell. For such lack of novelty and flaws in the computational methodology, I cannot recommend publication of this manuscript in its present form.

Response:

We thank the reviewer for this point and again apologize for the mistake in the computational details' explanation. As we mentioned above, indeed, for the for Pd₁ and Pd₂ calculations, we used 5×5 unit cell size with 4×4×1 in-plane k-point sampling. For our larger clusters (Pd₃ and Pd₄) we had used the 2×2×1 in-plane sampling for a 7×7 unit cell, which is reasonable for this large unit cell. To address the reviewer's concern, we have corrected the computational details and explained the details more carefully in the revised version of the SI. Also, for our 7×7 unit cell calculations we increased the k-point to 4×4×1, 6×6×1 and 8×8×1 (**Figure R3**) and calculated both the adsorption energy of the OOH* and the limiting potential on the Pd₃ cluster. As can be seen, the OOH* adsorption energy and the resultant limiting potentials remain the same for the higher k-point sampling, indicating that our previous results with 2×2×1 in-plane sampling for the 7×7 unit cell calculations are valid.

Figure R3. Top figure shows the optimized structure with the most stable coverage for Pd_3 cluster ($3\text{O}^*:1\text{HO}^*$) modeled in a 7×7 unit cell. (a) and (b) are the calculated adsorption energy of OOH^* and calculated limiting potential for different K-point sampling, respectively.

Action:

To address the reviewer's comment, we have corrected the computational details and explained the details more carefully in the methods by including the above figure as Supplementary Figure 36.

[Supplementary Figure 36. Top figure shows the optimized structure with the most stable coverage for Pd_3 cluster ($3\text{O}^*:1\text{HO}^*$) modeled in a 7×7 unit cell. (a) and (b) are the calculated adsorption energy of OOH^* and calculated limiting potential for different K-point sampling, respectively.]

[A 5×5 super cell lateral size was used to model Pd_1 and Pd_2 clusters and the Brillouin zone was sampled with $(4 \times 4 \times 1)$ Monkhorst-Pack k-points. For the larger clusters of Pd_3 and Pd_4 we used a 7×7 super cell lateral size with $(2 \times 2 \times 1)$ Monkhorst-Pack k-points sampling. Figure S36 displays the k-point sampling convergency plots for the calculated adsorption energies and

limiting potentials for an example of our model calculations (Pd₃ cluster covered with 3O* and 1HO*).]

Reviewer #2 (Remarks to the Author):

In this manuscript, the authors describe the preparation of partially oxidized Pd anchored on defect carbon. In recent years, the ORR performance of materials' in acidic conditions has been widely studied and the authors reported the performance of partially oxidized Pd with oxygen-functionalized carbon in promoting H₂O₂ production via 2 e- oxygen reduction in acidic electrolyte. They illustrated the synergistic effect of Pd and carbon and the materials showed nearly 100% selectivity toward H₂O₂ in a wide potential range of 0.3-0.7 V. This study is largely well carried out, and it is suitable to be published in Nat. Commun. However, some problems should be addressed before this work can be accepted for publication.

Response:

We thank the reviewer for his/her positive comments.

(1) In the abstract, the authors described the materials as partially oxidized palladium, however, there were no sufficient experimental evidences to prove that the palladiums were partially oxidized. Therefore, the authors should provide more evidences. And the materials were denoted as Pd-OCNT or H-Pd-OCNT. It maybe improper because the Pd atoms were partially oxidized.

Response:

We thank the reviewer for raising this important point. One strong evidence for partially oxidized Pd is the detection of the Pd-O bond in Pd-OCNT from the EXAFS analysis. This conclusion is also supported by the comparison of the X-ray absorption near edge spectra (XANES) of metallic Pd and Pd-OCNT, as shown in the Figure below (also added as Supplementary Figure 22 in the revised SI). The first K-edge feature of Pd-OCNT is more intense than that of metallic Pd, suggesting that Pd in Pd-OCNT is partially oxidized.

Figure R4. X-ray absorption near-edge structure (XANES) analysis of Pd^{δ+}-OCNT and Pd metal.

Action:

Based on reviewer's suggestion, we have changed our nomenclature from Pd-OCNT to Pd^{δ+}-OCNT to highlight the nature of partially oxidized Pd throughout the revised manuscript. **Figure R4** was also added in the supplementary information.

[Supplementary Figure. 22 X-ray absorption near-edge structure (XANES) analysis of Pd^{δ+}-OCNT and Pd metal.]

(2) In page 5, the authors stated that the surfaces become smooth after 8.5 h of oxidation due to the complete etching of outer-walls of OCNT. Why? Whether the authors studied the outer-surfaces' roughness when the oxidation time was over 8.5 h and whether the surfaces would become rough or not.

Response:

CNT could be oxidized with the assistance of nitric acid (HNO₃)²⁻³ or oxygen (O₂)⁴⁻⁵. For example, Ling et al.⁴ prepared OCNT by oxidizing the MWCNT in the presence of O₂ and examined the effect of oxidation time on the opening and thinning of MWNTs. With increased oxidation time, MWCNT was opened first (oxidation time = 30 min) and then became thinner

due to collapsed outer graphene layers. The similar peeling process was also observed by Cumings et al.⁶ using an electrically driven vaporization method. Therefore, we expect that the formation of OCNT via the nitric acid treatment should also follow the similar peeling mechanism, from the opening of outer surface to completely peeling it off to create a thinner OCNT with a smooth surface. The surface of OCNT with the oxidation time of 11 h was also examined by TEM. For 11 h OCNT, the mass loss was up to 70%. Such a large ratio of mass loss is associated with strong oxidation and etching effect on the entire wall of the CNTs. As shown below (**Figure R5**), not only outside graphene layers were peeled off in some parts of OCNT, but also some parts of the CNT were completely etched to the inner wall. Previous studies on the electrical properties of oxidized CNTs have shown that strong oxidation can result in significantly decreased electrical conductivity (e.g., Špitalský, Zdenko, et al. *Compos. Part A Appl. Sci. Manuf.* (2009): 778-783.; Kim, Yoon Jin, et al. *Carbon* (2005): 23-30.)⁷⁻⁸ due to similar structure defects, which is consistent with our results.

Figure R5. HRTEM image of obtained OCNT with the oxidation time of 11 h.

(3) In page 5, the authors stated that the increase of pseudocapacitive current as the oxidation time extended from 2.5 h to 8.5 h could be attributed to the increased number of C-O groups. However, in the XPS results, as stated in the 144 line in page 5, the ratio of C-O groups was

decreased by ~5% when the oxidation time was 8.5 h. So the explanation is not consistent with the XPS results. The authors should think over the results carefully.

Response:

Based on the reviewer's suggestion we calculated the capacitances from the pseudocapacitive current for different OCNT samples and found that the trend of capacitances from the pseudocapacitive current generally follows that of the relative ratio of the C-O groups. It thus suggested that redox peaks were attributed to the oxidation/reduction of surface quinoidal functional groups⁹. The possible reason for the slightly improved capacitance (by 0.5%) of 8.5 h OCNT compared with 6 hr OCNT might be the increased electrochemical surface area accessible to the electrolyte with extended oxidation time, which compensates the reduced relative ratio of the C-O groups.

Figure R6. Calculated relative ratios of C-O groups and capacitance of redox peaks from CV for different OCNT samples.

Action:

We revised the discussion as below and added the figure in Supplementary Figure.13.

[As shown in the CV curves (Supplementary Figure. 13A), the pseudocapacitive current (represented by the redox peaks between 0.2 and 0.8 V) of the OCNT electrodes first increased

as the oxidation time extended from 2.5 to 6.5 h and then maintained roughly the same from 6.5 to 8.5 h. The trend of capacitance from the redox current was similar with that of the relatively ratios of the C-O groups on the surface from the XPS results (Supplementary Figure.13B), suggesting that redox peaks could be attributed to the oxidation/reduction of surface quinoidal functional groups³¹.

[Supplementary Figure 13. (A) Cyclic voltammogram curves of OCNT samples in Ar-saturated 0.1 M HClO₄ with a scan rate of 50 mV s⁻¹. (B) Calculated relative ratios of C-O groups and the capacitance of redox peaks from CV for different OCNT samples.

The possible reason for the slightly improved capacitance (by 0.5%) of 8.5 h OCNT compared with 6.5 h OCNT might be the increased electrochemical surface area accessible to the electrolyte.]

(4) The captions of supplementary Fig 4 and 24. are not appropriate. For example, the caption of supplementary Fig 4. should be changed as follows: The N₂ adsorption-desorption isotherms of OCNT samples.

Response:

We thank the reviewer for the comment.

Action:

The captions of supplementary Fig. 4 and Fig.24 were changed as suggested.

[Supplementary Figure 4. The N₂ adsorption-desorption isotherms of OCNT samples.]

[Supplementary Figure 25. The N₂ adsorption-desorption isotherms of Pd^{δ+}-OCNT and H-Pd-OCNT.]

Reviewer #3 (Remarks to the Author):

The authors present an interest report into the partially oxidised Pd on CNTs for selective production of H₂O₂ via oxygen reduction in acidic media. The motivations for the work are clear, and the case for Pd-O-C as an efficient catalyst unit is strong and supported by experimental and computational data. The clear distinction between the Pd cluster and larger Pd particles is well defined and supported. In particular, it is encouraging to see the catalyst performing well both under RRDE and H-cell operating conditions. I would like to raise a couple of points that I think will strengthen the paper, but beyond these I am happy to recommend this article for publication.

Response:

We thank the reviewer for his/her careful review and positive comments.

(1) Line 89: The authors introduce the phrase "optimum *OOH binding energy" before any discussion of ORR mechanism, intermediates or DFT. It may be worth replacing this with a more general description to aid the more general interest reader.

Response:

We thank the review for the useful comment.

Action:

[Line 89: “Benefiting from its optimal HOO* binding energy,” was changed to “Benefiting from its optimal binding energy for the key intermediate (HOO*) in 2e⁻ ORR,”]

(2) Line 72: The authors may wish to include B- or N-doped carbons in their discussions of defective carbon materials for H₂O₂ from ORR.

Response:

We agree with reviewer’s suggestion.

Action:

The discussion for B-doped carbon and N-doped carbons were both added.

[In alkaline and neutral electrolytes, defective carbon materials, such as oxidized carbon nanotubes (O-CNT)¹⁰, B-N-doped carbon¹¹, Fe single-atom coordinated O-CNT¹² and reduced graphene oxide (GO)¹³, have shown high activity and selectivity for the 2e⁻ ORR. For example, mildly reduced GO exhibit nearly 100% selectivity and stable activity at low overpotential (< 10 mV) in the 0.1M KOH¹³⁻¹⁴. It is particularly interesting to find that the selectivity of carbon materials could also be enhanced by the introduction of boron nitride (BN) islands where the active sites were attributed to the interface between hexagonal BN and graphene.¹¹ While these catalysts are efficient in alkaline conditions, producing H₂O₂ under acidic conditions shows technological advantage in fuel cell operation as today’s proton conducting polymeric membranes are far more technologically mature than their hydroxide-conducting counterparts¹⁵⁻¹⁶. In addition, acidic- H₂O₂ solution can be directly used as an oxidizer for chemical synthesis, which attributes to more than 33% of the global market share of H₂O₂¹⁷⁻¹⁸. Due to the weak acidic nature of the H₂O₂ molecule,¹⁹ storing H₂O₂ in an acidic environmental can also offer a longer shelf-life compared to alkaline conditions. However, carbon-based materials require a large overpotential (~300 mV) to initiate the ORR reaction in acidic electrolytes, resulting in significant voltage loss in fuel cell operations²⁰⁻²¹. For instance, the onset potential of high-selectivity mesoporous N-doped carbon was up to ~0.5 V in 0.1 M HClO₄, leading to a possible potential loss of 200 mV in the ORR test.²⁰]

(3) Figures throughout: Unit convention for potential axes should keep V as the unit alone, i.e. E

vs RHE (V) rather than E (V vs RHE)

Response:

We thank the reviewer for the suggestion.

Action:

The unit convention for potential axes in all figures were changed to E vs. RHE (V).

[Fig. 3]

[Supplementary Figure 13, 14, 15, 16, 17, 18, 26, 27]

(4) Figure 3B: The current density for H-Pd-OCNT is significantly lower than for Pd-OCNT, though the authors state that the ORR is switching from the 2e⁻ to the 4e⁻ mechanism. This seems counter-intuitive, as moving to the 4e⁻ mechanism should cause a larger magnitude current to pass. Do the authors have an explanation for this hindered current.

Response:

After heat treatment, ORR activities of H-OCNT and H-Pd-OCNT electrocatalysts both decreased due to the change of functional groups as shown from XPS. Both 6.5 h OCNT and 6.5h H-OCNT present the 2e⁻ ORR reaction pathway with a similar H₂O₂ selectivity, but the current density of H-OCNT at 0.1 V (~1.2 mA cm⁻²) was much lower than that of OCNT (~2.4 mA cm⁻²), suggesting a decreased ORR activity of H-OCNT after heat treatment (Supplementary Figure. 18 and 27). For electrocatalysts with low ORR activities, the lower magnitude of current density for 4e⁻ ORR was also common in previous reports.²²⁻²⁴ Therefore, the hindered current could be ascribed to the poor 4e⁻ ORR activity of H-Pd-OCNT after heat treatment.

(5) Supplementary Figure 18: The triangle and circle symbols in the legend appear to be the wrong way around

Response:

We thank the reviewer for the careful review.

Action:

The legends in the Supplementary Figure 18 were revised.

[Supplementary Figure 18. RRDE voltammograms in O_2 -saturated HClO_4 electrolyte with a scan rate of 10 mV s^{-1} at 1600 rpm (only the anodic cycle is shown). The disc current, ring current, hydrogen peroxide current calculated from the ring current and selectivity during the reaction of OCNT samples reacted from (A) 2.5 h, (B) 4.5 h, (C) 6.5 h, (D) 8.5 h.]

(6) Supplementary Figure 19: If the bars stay unattached to an axis, it would be useful to add data labels, or some other way of indicating the magnitude of the selectivity change with reaction time

Response:

We thank the reviewer for the careful review.

Action:

The data labels were added in the Supplementary Figure. 19.

[Supplementary Figure 19. Summary of the relationship between the ratios of defects, C-O groups in OCNT samples reacted from 2.5 h to 8.5 h and its selectivity toward 2e⁻ ORR.]

(7) Supplementary Figure 26: Legend is missing

Response:

We thank the reviewer for the careful review.

Action:

The legend was added in the Supplementary Figure. 27.

[Supplementary Figure 27. RRDE voltammograms in O_2 -saturated HClO_4 electrolyte with a scan rate of 10 mV s^{-1} at 1600 rpm (only the anodic cycle is shown). The disc current, ring current, hydrogen peroxide current calculated from the ring current and selectivity during the reaction of H-OCNT sample.]

(8) Line 246: It would be useful to discuss the operation of the H-cell in terms of current density and to give an idea of the H-cell volume and end concentration of H_2O_2 to give a clear picture of peroxide production rate. Similar scaling information would be useful for supplementary figure 27 as well.

Response:

We thank the reviewer for the valuable comment.

The total volume of H-cell (half part) is 30 ml. In the test, 17 ml of electrolyte was added into the working electrode chamber. The nominal current densities of the H-cell could be adjusted by changing the mass loading of the electrocatalysts in the working electrode. When 50 μl of electrocatalyst ink (corresponding to a catalyst mass loading of 0.1 mg/cm^2) was deposited on the working electrode with an area of $\sim 0.49 \text{ cm}^2$, the current density was 10 mA cm^{-2} at 0.1 V , which could produce H_2O_2 with a concentration of 10 wt%, close to the optimal concentration of dilute H_2O_2 solution (9 wt%) used in chemical synthesis (acid-based). The scaling information for the H-cell was added in the Supplementary Figure.27.

Action:

The discussion for the current density in H-cell test and the scaling information for the H-cell were both added. The scaling information was added in the counterpart (the same size with the working electrode part) of H-cell.

[When the catalyst mass loading was controlled to 0.1 mg cm^{-2} , a steady current density of 10 mA cm^{-2} was recorded at 0.1 V (Fig. 4A). The selectivity of H_2O_2 was measured to be 87%, which was close to the RRDE test at 0.1 V . Also, the yield of H_2O_2 was up to $1701 \text{ mol kg}_{\text{cat}}^{-1} \text{ h}^{-1}$, 2 times higher than that of the single atomic Pt electrocatalyst reported recently.²⁵ Most importantly, the end H_2O_2 concentration reached 10 wt% after 35 min of operation, which could be readily used for acid-based chemical synthesis (9 wt% is commonly used).¹⁶]

[Supplementary Figure 28. The digital photo image of the H-cell used for the H₂O₂ yield test. The electrolyte volume in the working electrode part was 17 ml and the total volume of this part was 30 ml.]

(9) Line 247: The titration method appears to be missing from the supplementary information. Related to this section, does the titration method take H₂O₂ decomposition into account during the H-cell operation?

Response:

The titration method was in Line 93-99 in previous supplementary information (now in the method). For the titration method, the H₂O₂ decomposition was not taken into account during the H-cell operation. The amount of H₂O₂ was calibrated by the concentration of Ce⁴⁺ from the reaction ($2\text{Ce}^{4+} + \text{H}_2\text{O}_2 \rightarrow 2\text{Ce}^{3+} + 2\text{H}^+ + \text{O}_2$) and the concentration of Ce⁴⁺ after the reaction was measured by ultraviolet–visible spectroscopy with 316 nm of wavelength. By comparing the selectivity result from the titration method (87%) with RRDE (91%), the difference was only 4% due to the decomposition of H₂O₂. Therefore, we think that the titration method used for the calculation of H₂O₂ selectivity is acceptable. Also, this method was widely used in previous studies, such as *Nat Catal* 1, 156-162 (2018)¹⁰, *Adv. Energy Mater.*, 8, 1801909 (2018)¹⁷.

(10) Similarly, the selectivity of 87% at 5 mA is promising for higher production rates of H₂O₂. Do the authors have any indication of how this material would perform under higher operating conditions ~100 mA cm⁻²? Is this material a candidate for replacing the anthraquinone process? Specifically, I wonder how stable the partially oxidised environment will be under more harshly reducing conditions.

Response:

The current density could be further enhanced by increasing the electrocatalyst mass loading on the working electrode. Based on the reviewer's suggestion, we performed additional H-cell experiments with increased electrocatalyst mass loadings. Stable current densities of 19 and 55 mA cm⁻² could be obtained when the electrocatalyst mass loadings were increased to 0.2 and 0.6 mg cm⁻², respectively. When the mass loading increased to 1.2 mg cm⁻², the thick catalyst layer (catalysts were deposited on a relatively small area of ~0.49 cm⁻² due to the size limit of the chamber) cracked more easily and the catalysts tended to peel off from the electrode during the

test due to the large O₂ flux, resulting in current density decay from 100 to 78 mA cm⁻² during the 1 h operation. In previous studies, the highest current density in the H-cell test was 20 mA cm⁻² (Table R4). In future studies, we will explore flow membrane electrode assemble (MEA) to investigate the performance under the higher operating conditions for industrial applications, such as 100 mA cm⁻².

Table R1. Recent studies of current densities obtained from H-cell test

Electrocatalyst	Current density (mA cm ⁻²)	Reference
Pd ^{δ+} -OCNT	55	Our work
Reduced graphene oxide	1.5 mA	13
Oxidized carbon nanotubes	20	10
Single Pt atom@CuS _x	~8	25

The oxygen reduction reaction (ORR) is usually performed in a potential range of 1.05 – 0.05 V. The lower potential, the harsher the reducing condition is. For the reaction at 0.05 V, the small background current (CV current) was notable (See Supplementary Figure.13A). So, we performed the stability test at 0.1 V, which was nearly the harshest reducing condition for a H-cell test. We also performed the stability test for higher current density in an H-cell test with the mass loading of 0.6 mg cm⁻². The current density showed a small decay (~15%) after 24000 s.

Figure R7. Stability test in H-cell. (A) Current densities obtained from H-cell test by adjusting the mass loadings of electrocatalyst at 0.1 V. (B) H-cell stability test with a electrocatalyst mass loading of 0.6 mg cm^{-2} at 0.1 V.

The purpose of electrochemical H_2O_2 production is to directly produce dilute H_2O_2 solution on-site. For the H_2O_2 applications required dilute H_2O_2 solution, such as chemical synthesis, we believe that the current electrochemical synthesis method can be a promising candidate for replacing the anthraquinone process. As we pointed our earlier, in future studies, we will explore flow MEA setup to directly produce dilute H_2O_2 solution at large operation current (e.g., 100 mA/cm^2) for the demonstration of industrial applications.

Action:

The discussion and **Figure R7** were added in the supplementary information.

[Supplementary Figure 29. Stability test in H-cell. (A) Current densities obtained from H-cell test by adjusting the mass loadings of electrocatalyst at 0.1 V. (B) H-cell stability test with a electrocatalyst mass loading of 0.6 mg cm^{-2} at 0.1 V.]

We performed additional H-cell experiments with increased electrocatalyst mass loadings. Stable current densities of 19 and 55 mA cm^{-2} could be obtained when the electrocatalyst mass loadings were increased to 0.2 and 0.6 mg cm^{-2} , respectively. When the mass loading increased to 1.2 mg cm^{-2} , the thick catalyst layer (catalysts were deposited on a relatively small area of $\sim 0.49 \text{ cm}^2$ due to the size limit of the chamber) cracked more easily and the catalysts tended to feel off the electrode during the test due to the large O_2 flux, resulting in current density decay from 100 to 78 mA cm^{-2} during the 1 h operation.]

(11) Figure 5: The interaction with the epoxy unit gives a clear shift towards the top of the volcano. Can the authors comment on how susceptible ΔG_{HOO^*} is to the nature of this surface bound oxygen? Presumably the OCNTs will show a range of different surface bound O?

Response:

We thank the reviewer for this point and would like to mention that we have explored the effect of a range of different oxygen functional groups including hydroxyl, carbonyl and etheric groups. Among all these functional groups, we found epoxy groups to have the most meaningful impact on the ΔG_{HOO^*} which explained the experimental results. The figure below displays the activity volcano including an example of the effect of hydroxyl group on the adsorption energy of HOO^* and calculated limiting potential for Pd_3 $3\text{O}^*:\text{HO}^*$. This figure shows that similar to epoxy group, including the hydroxyl functional group weakens the adsorption energy of HOO^* . This in turn results in increasing the selectivity toward H_2O_2 ,²⁶ however, the calculated limiting potential is not significantly high.

Figure R8. (a) Optimized DFT model structures with nearby hydroxyl functional groups (highlighted by orange dashed circles) on the most stable O^*/HO^* covered Pd_3 clusters. (b) Activity volcano plot, including the hydroxyl functional group for the example of Pd_3 cluster.

Action:

To Address reviewer's comment, we added these details and Figure in the supplementary information page 39 and Figure 38.

[The effect of other oxygen functional groups:]

We have explored the effect of a range of different oxygen functional groups including hydroxyl, carbonyl and etheric groups. Among all these functional groups, we found epoxy groups to have

the most meaningful impact on the ΔG_{HOO^*} which well aligned with the experimental results. Supplementary Fig.38 displays the activity volcano including an example of the effect of hydroxyl group on the adsorption energy of HOO^* and calculated limiting potential for Pd_3 covered with $3\text{O}^*:\text{HO}^*$. The results show that similar to epoxy group, including the hydroxyl functional group weakens the adsorption energy of HOO^* . This in turn results in increasing the selectivity toward H_2O_2 ,²⁶ however, the calculated limiting potential is not significantly high.]

[Supplementary Fig.38 (a) Optimized DFT model structures with nearby hydroxyl functional groups (highlighted by orange dashed circles) on the most stable O^*/HO^* covered Pd_3 clusters. (b) Activity volcano plot, including the hydroxyl functional group for the example of Pd_3 cluster.]

Reference

1. Siahrostami, S.; Verdaguer-Casadevall, A.; Karamad, M.; Deiana, D.; Malacrida, P.; Wickman, B.; Escudero-Escribano, M.; Paoli, E. A.; Frydendal, R.; Hansen, T. W., Enabling direct H_2O_2 production through rational electrocatalyst design. *Nat. Mater.* **2013**, *12* (12), 1137.
2. Rosca, I. D.; Watari, F.; Uo, M.; Akasaka, T., Oxidation of multiwalled carbon nanotubes by nitric acid. *Carbon* **2005**, *43* (15), 3124-3131.
3. Avilés, F.; Cauich-Rodríguez, J.; Moo-Tah, L.; May-Pat, A.; Vargas-Coronado, R., Evaluation of mild acid oxidation treatments for MWCNT functionalization. *Carbon* **2009**, *47* (13), 2970-2975.
4. Chang, J.-Y.; Ghule, A.; Chang, J.-J.; Tzing, S.-H.; Ling, Y.-C., Opening and thinning of multiwall carbon nanotubes in supercritical water. *Chem. Phys. Lett.* **2002**, *363* (5-6), 583-590.
5. Frank, B.; Rinaldi, A.; Blume, R.; Schlögl, R.; Su, D. S., Oxidation stability of multiwalled carbon nanotubes for catalytic applications. *Chem. Mater.* **2010**, *22* (15), 4462-4470.
6. Cumings, J.; Collins, P. G.; Zettl, A., Peeling and sharpening multiwall nanotubes. *Nature* **2000**, *406* (6796), 586-586.
7. Špitalský, Z.; Krontiras, C. A.; Georga, S. N.; Galiotis, C., Effect of oxidation treatment of multiwalled carbon nanotubes on the mechanical and electrical properties of their epoxy composites. *Composites Part A: Applied Science and Manufacturing* **2009**, *40* (6-7), 778-783.
8. Kim, Y. J.; Shin, T. S.; Do Choi, H.; Kwon, J. H.; Chung, Y.-C.; Yoon, H. G., Electrical conductivity of chemically modified multiwalled carbon nanotube/epoxy composites. *Carbon* **2005**, *43* (1), 23-30.
9. Ye, J.-S.; Liu, X.; Cui, H. F.; Zhang, W.-D.; Sheu, F.-S.; Lim, T. M., Electrochemical oxidation of multi-walled carbon nanotubes and its application to electrochemical double layer capacitors. *Electrochem. Commun.* **2005**, *7* (3), 249-255.
10. Lu, Z.; Chen, G.; Siahrostami, S.; Chen, Z.; Liu, K.; Xie, J.; Liao, L.; Wu, T.; Lin, D.; Liu, Y., High-efficiency oxygen reduction to hydrogen peroxide catalysed by oxidized carbon materials. *Nature Catalysis* **2018**, *1* (2), 156.

11. Chen, S.; Chen, Z.; Siahrostami, S.; Higgins, D.; Nordlund, D.; Sokaras, D.; Kim, T. R.; Liu, Y.; Yan, X.; Nilsson, E., Designing boron nitride islands in carbon materials for efficient electrochemical synthesis of hydrogen peroxide. *J. Am. Chem. Soc.* **2018**, *140* (25), 7851-7859.
12. Jiang, K.; Back, S.; Akey, A. J.; Xia, C.; Hu, Y.; Liang, W.; Schaak, D.; Stavitski, E.; Nørskov, J. K.; Siahrostami, S., Highly selective oxygen reduction to hydrogen peroxide on transition metal single atom coordination. *Nat. Commun.* **2019**, *10* (1), 1-11.
13. Kim, H. W.; Ross, M. B.; Kornienko, N.; Zhang, L.; Guo, J.; Yang, P.; McCloskey, B. D., Efficient hydrogen peroxide generation using reduced graphene oxide-based oxygen reduction electrocatalysts. *Nature Catalysis* **2018**, *1* (4), 282.
14. Verdager-Casadevall, A.; Deiana, D.; Karamad, M.; Siahrostami, S.; Malacrida, P.; Hansen, T. W.; Rossmeisl, J.; Chorkendorff, I.; Stephens, I. E., Trends in the electrochemical synthesis of H₂O₂: enhancing activity and selectivity by electrocatalytic site engineering. *Nano Lett.* **2014**, *14* (3), 1603-1608.
15. Varcoe, J. R.; Atanassov, P.; Dekel, D. R.; Herring, A. M.; Hickner, M. A.; Kohl, P. A.; Kucernak, A. R.; Mustain, W. E.; Nijmeijer, K.; Scott, K., Anion-exchange membranes in electrochemical energy systems. *Energy & environmental science* **2014**, *7* (10), 3135-3191.
16. Yang, S.; Verdager-Casadevall, A.; Arnarson, L.; Silvioli, L.; Čolić, V.; Frydendal, R.; Rossmeisl, J.; Chorkendorff, I.; Stephens, I. E., Toward the decentralized electrochemical production of H₂O₂: a focus on the catalysis. *ACS Catalysis* **2018**, *8* (5), 4064-4081.
17. Jiang, Y.; Ni, P.; Chen, C.; Lu, Y.; Yang, P.; Kong, B.; Fisher, A.; Wang, X., Selective Electrochemical H₂O₂ Production through Two-Electron Oxygen Electrochemistry. *Advanced Energy Materials* **2018**, *8* (31), 1801909.
18. <https://www.gminsights.com/industry-analysis/hydrogen-peroxide-market>.
19. Gibian, M. J.; Elliott, D. L.; Hardy, W. R., Reaction of a sulfonyl-chymotrypsin with hydrogen peroxide. Generation of a hydroperoxy enzyme. *J. Am. Chem. Soc.* **1969**, *91* (26), 7528-7530.
20. Fellingner, T.-P.; Hasché, F. d. r.; Strasser, P.; Antonietti, M., Mesoporous nitrogen-doped carbon for the electrocatalytic synthesis of hydrogen peroxide. *J. Am. Chem. Soc.* **2012**, *134* (9), 4072-4075.
21. Park, J.; Nabae, Y.; Hayakawa, T.; Kakimoto, M.-a., Highly selective two-electron oxygen reduction catalyzed by mesoporous nitrogen-doped carbon. *ACS Catalysis* **2014**, *4* (10), 3749-3754.
22. Chang, Y.; Yuan, C.; Liu, C.; Mao, J.; Li, Y.; Wu, H.; Wu, Y.; Xu, Y.; Zeng, B.; Dai, L., B, N co-doped carbon from cross-linking induced self-organization of boronate polymer for supercapacitor and oxygen reduction reaction. *J. Power Sources* **2017**, *365*, 354-361.
23. Kramm, U. I.; Herrmann-Geppert, I.; Behrends, J.; Lips, K.; Fiechter, S.; Bogdanoff, P., On an easy way to prepare metal–nitrogen doped carbon with exclusive presence of MeN₄-type sites active for the ORR. *J. Am. Chem. Soc.* **2016**, *138* (2), 635-640.
24. Feng, Y.; He, T.; Alonso-Vante, N., In situ free-surfactant synthesis and ORR-electrochemistry of carbon-supported Co₃S₄ and CoSe₂ nanoparticles. *Chem. Mater.* **2008**, *20* (1), 26-28.
25. Shen, R.; Chen, W.; Peng, Q.; Lu, S.; Zheng, L.; Cao, X.; Wang, Y.; Zhu, W.; Zhang, J.; Zhuang, Z., High-Concentration Single Atomic Pt Sites on Hollow Cu_xS for Selective O₂ Reduction to H₂O₂ in Acid Solution. *Chem* **2019**.

26. Chen, S.; Chen, Z.; Siahrostami, S.; Kim, T. R.; Nordlund, D.; Sokaras, D.; Nowak, S.; To, J. W.; Higgins, D.; Sinclair, R., Defective carbon-based materials for the electrochemical synthesis of hydrogen peroxide. *ACS Sustainable Chemistry & Engineering* **2018**, 6 (1), 311-317.

Reviewers' Comments:

Reviewer #1:

Remarks to the Author:

The authors have acknowledged that computational models were not properly described. It has been updated and, when needed, they performed additional calculations to ensure the strength of their numerical parameters. I think that the computational part is now much more reliable than before and I am happy now to recommend this manuscript for publication

Reviewer #2:

Remarks to the Author:

I think the authors have addressed my previous comments and the revised version can be accepted now.

Reviewer #3:

Remarks to the Author:

The authors have carefully considered all comments I made in my review. Their responses are convincing and well supported with literature and experimental evidence. I have no further suggestions for this paper and am happy to recommend for publication.